

# Cyclin-dependent kinase inhibitor 1 plays a more prominent role than activating transcription factor 4 or the p53 tumour suppressor in thapsigargin-induced G1 arrest

Erin van Zyl[1], Claire Peneycad[1], Evan Perehiniak[1] and Bruce C. McKay[1,2]

[1] Department of Biology, Carleton University, Ottawa, ON, Canada
[2] Institute of Biochemistry, Carleton University, Ottawa, ON, Canada

Corresponding author
Bruce C. McKay,
bruce_mckay@carleton.ca

## ABSTRACT

**Background**. Thapsigargin (Tg) is a compound that inhibits the SERCA calcium transporter leading to decreased endoplasmic reticulum (ER) Ca2+ levels. Many ER chaperones are required for proper folding of membrane-associated and secreted proteins, and they are Ca2+ dependent. Therefore, Tg leads to the accumulation of misfolded proteins in the ER, activating the unfolded protein response (UPR) to help restore homeostasis. Tg reportedly induces cell cycle arrest and apoptosis in many cell types but how these changes are linked to the UPR remains unclear. The activating transcription factor 4 (ATF4) plays a key role in regulating ER stress-induced gene expression so we sought to determine if ATF4 is required for Tg-induced cell cycle arrest and apoptosis using ATF4-deficient cells.

**Methods**. Two-parameter flow cytometric analysis of DNA replication and DNA content was used to assess the effects of Tg on cell cycle distribution in isogenic HCT116-derived cell lines either expressing or lacking ATF4. For comparison, we similarly assessed the Tg response in isogenic cell lines deleted of the p53 tumour suppressor and the p53-regulated p21$^{WAF1}$ cyclin-dependent kinase inhibitor important in $G_1$ and $G_2$ arrests induced by DNA damage.

**Results**. Tg led to a large depletion of the S phase population with a prominent increase in the proportion of HCT116 cells in the $G_1$ phase of the cell cycle. Importantly, this effect was largely independent of ATF4. We found that loss of p21$^{WAF1}$ but not p53 permitted Tg treated cells to enter S phase and synthesize DNA. Therefore, p21$^{WAF1}$ plays an important role in these Tg-induced cell cycle alterations while ATF4 and p53 do not. Remarkably, the ATF4-, p53-and p21$^{WAF1}$-deficient cell lines were all more sensitive to Tg-induced apoptosis. Taken together, p21$^{WAF1}$ plays a larger role in regulating Tg-induced $G_1$ and $G_2$ arrests than ATF4 or p53 but these proteins similarly contribute to protection from Tg-induced apoptosis. This work highlights the complex network of stress responses that are activated in response to ER stress.

## INTRODUCTION

The endoplasmic reticulum (ER) is a large organelle that has a variety of important functions. Importantly, the ER is the primary site for calcium (Ca2+) storage in the cell and translation of membrane-associated and secreted proteins occurs in ribosomes associated with the ER. The lumen of the ER is rich in chaperone proteins and Ca2+ is an important cofactor for some ER chaperones, including glucose regulated proteins (Grp94 and Grp78) (*Coe & Michalak, 2009*). Alterations in the concentration of free Ca2+ in the lumen of the ER has pronounced effects on the capacity of chaperones to fold proteins (*Carreras-Sureda, Pihán & Hetz, 2018*). Thapsigargin (Tg) is a competitive inhibitor of the sarco endoplasmic reticulum Ca2+ ATPase (SERCA) that pumps Ca2+ from the cytosol into the lumen of the ER. Therefore, Tg decreases ER Ca2+ levels while increasing cytosolic Ca2+. The decrease in ER Ca2+ concentration decreases chaperone activity and leads to the accumulation of misfolded proteins causing ER stress. This form of ER stress activates the unfolded protein response (UPR). Three ER transmembrane proteins (Ire1, ATF6 and PERK) are activated in the UPR and each functions in a distinct manner to elicit a downstream transcriptional response that facilitates recovery from ER stress (*Walter & Ron, 2011*).

PERK is a kinase that along with a group of three additional stress-responsive kinases, PKR, HRI and GCN1, transmit stress responses through a common cascade that is known as the integrated stress response (ISR). These four kinases phosphorylate the translation initiation factor eIF2α to down regulate cap-dependent translation. Despite a decrease in global translation, several transcripts are selectively translated including activation transcription factor 4 (ATF4) (*Vattem & Wek, 2004*). This protein is then translocated to the nucleus where it functions as a transcription factor in response to diverse stresses. We recently created ATF4-deficient HCT116 cells (ATF4-def), a useful model for investigating the role of this transcription factor in the response to a variety of cellular stresses (*van Zyl et al., 2021*).

Tg, and more generally ER stress, induces PERK-dependent alterations in the cell cycle at a variety of stages (*Han et al., 2013*; *Hamanaka et al., 2005*; *Cabrera et al., 2017*; *Popat, Patel & Warnes, 2019*). Short term exposure (2 h) to Tg can inhibit DNA synthesis leading to S phase arrest in a PERK-dependent manner (*Cabrera et al., 2017*). Longer ER stress (*i.e.* 20 h) elicited in a variety of ways (*i.e.* tunicamycin or low glucose) can lead to $G_1$ arrest in a PERK- and GCN2-dependent manner (*Hamanaka et al., 2005*). However, it was unclear whether the downstream ATF4 transcription factor plays a central role in cell cycle alteration in response ER stress. $G_2$ arrest can also be induced by Tg in a PERK-dependent manner that appears to involve an isoform of the p53 tumour suppressor (*Bourougaa et al., 2010*). Forced expression of ATF4 induces $G_1$ arrest (*Frank et al., 2010*). Both ATF4 and p53 can regulate expression of the cyclin-dependent kinase inhibitor p21$^{WAF1}$ under some conditions (*Inoue et al., 2017*; *Ebert et al., 2020*; *Lehman et al., 2015*).

Here we used isogenic sets of HCT116-derived cell lines deleted of ATF4, the p53 tumour suppressor or the cyclin dependent kinase inhibitor p21$^{WAF1}$ to assess the relative

contribution of these proteins to cell cycle alterations following Tg exposure. Using two-parameter flow cytometric analysis of BrdU incorporation and DNA content, we found that Tg-treated HCT116 cells accumulated in either $G_1$ or $G_2$ with an almost complete depletion of the BrdU-positive S phase population over a broad range of concentrations. In ATF4- and p53-deficient sublines of HCT116 cells, Tg also induced strong $G_1$ arrest with S phase depletion, but these cell cycle changes were somewhat less pronounced. In contrast, p21$^{WAF1}$ null HCT116 cells continued to cycle in the presence of Tg indicating that $G_1$ and $G_2$ arrests were p21$^{WAF1}$-dependent. Despite the differential contribution of p21$^{WAF1}$, p53 and ATF4 to $G_1$ and $G_2$ arrests, these proteins appeared to be similarly effective at preventing apoptosis across all Tg concentrations. These results suggest that ATF4, p53 and p21$^{WAF1}$ play adaptive roles in response to Tg exposure, but this is unlikely to be through a common mechanism involving cell cycle checkpoints.

## MATERIALS & METHODS

### Cell Culture and drug treatments

HCT116 cells and sublines that carry deletions in the *ATF4*, *TP53* or *CDKN1A* genes (referred to as ATF4-def, p53-null and p21-null, respectively) (*van Zyl et al., 2021*; *Bunz et al., 1998*) were grown in McCoys media (Hyclone, San Angelo, TX, USA) supplemented with a 12% mixture of Newborn Calf Serum and Fetal Bovine Serum (3:1) ratio (Gibco, ThermoFisher Scientific, Ottawa, Canada) and 90 units/ml penicillin, 90 μg/ml streptomycin (Hyclone). The p53-null, p21-null and their parental HCT116 strain were obtained from Dr. Bert Vogelstein (Johns Hopkins University) (*van Zyl et al., 2021*; *Bunz et al., 1998*). The ATF4-def cells were derived from HCT116 cell line obtained directly from the American Type Culture Collection (Cat #: CCL-247; Manassa, VA, USA) (*van Zyl et al., 2021*). Normal human neonatal foreskin fibroblasts expressing human telomerase (NFhTert) were obtained from Mats Ljungman (University of Michigan) (*O'Hagan & Ljungman, 2004*). These cells were maintained in DMEM supplemented with 10% fetal bovine serum and 90 units/ml penicillin, 90 μg/ml streptomycin (Hyclone). In all experiments, cells were seeded 24 h prior to treatment at 500,000 cells/well of a 6-well plate or 100,000 cells/12-well plate. Cells were left untreated, treated with up to 2 μM Tg (Millipore Sigma Canada, Oakville ON, Canada), or treated with a volume of dimethyl sulfoxide (DMSO) (Calbiochem) equal in volume to the highest Tg treatment as a vehicle control. Where indicated, 25 μM zVAD-fmk (Millipore Sigma Canada), was added alone or in combination with 2 μM Tg to block caspase activity.

### Immunoblotting

Immunoblotting was completed as described previously (*van Zyl et al., 2021*). Briefly, media was removed, and cells were washed with phosphate buffered saline (PBS) pH 7.4. Cells monolayers were collected in RIPA buffer. Samples were sonicated and quantified using the Bio-Rad protein assay (Bio-Rad, Hercules, CA, USA). Samples were denatured in NuPAGE™ LDS Sample Buffer and NuPAGE™ Sample Reducing Agent (10X) (Thermo Fisher Scientific, Ottawa, Canada), heated for 10 min at 70 °C and equal amounts of protein were loaded onto NuPAGE 4–12% Bis-Tris gels (Thermo Fisher Scientific). The

protein was transferred to a nitrocellulose membrane (Bio-Rad). 1mg/ml Ponceau S in 1% acetic acid was used to detect the transferred protein. The membrane was then blocked in 5% milk in TBST pH 7.6 for a minimum of 1 h at room temperature. The membrane was then incubated in primary antibody overnight at 4 °C. All primary antibodies were prepared in 0.5% milk in TBST at 1:200 for anti-p21WAF1 (Ab-1, Millipore Sigma) and anti-p53 (sc-6243, Santa Cruz) and 1:1,000 for anti-ATF4 (abcam) and anti-beta-Actin (A5316, Sigma Aldrich). The membrane was then washed 4 times for 5 min with TBST, and incubated in secondary antibody (1:10,000 for HRP-conjugated goat anti-mouse and 1:20,000 for HRP-conjugated goat anti-rabbit; Abcam, Cambridge, UK) for 2–3 h at room temperature. The membrane was washed 4 times with TBST for 5 min, followed by a 10-minute wash. Clarity™ Western ECL substrate (Bio-Rad) was added, and the membrane was imaged using the Fusion FX5 gel documentation system (Vilber Lourmat, Collégien, France). To visualize additional proteins, the membrane was subsequently washed 4 times in TBST, blocked in 5% milk in TBST again and incubated in primary antibody, as described above.

## Quantitative reverse transcriptase polymerase chain reaction (qRT-PCR)

Medium was removed and cells were rinsed twice with PBS (pH 7.4). RNA was extracted using the EZ-10 DNA away RNA Miniprep kit (Bio Basic Canada Inc, Markham, Canada). RNA concentration and quality was measured using a DeNovix DS-11 Spectrophotometer (DeNovix, Wilmington, DE, USA). Equal amounts of RNA were converted to cDNA using the High-Capacity cDNA Reverse Transcription Kit (Thermo Fisher Scientific, Ottawa Canada). qPCR was performed using Bioline SensiFAST™ Probe HI-ROX Master Mix (FroggaBio Inc., Toronto, ON) with SYBR green. The primers used were obtained from integrated DNA technologies (IDT) and the sequences were CDKN1A (GGAGACTCTCAGGGTCGAAA, and GCTTCCTCTTGGAGAAGATCAG), and GAPDH (AGCCACATCGCTCAGACA, and GCCCAATACGACCAAATC).

## Cell cycle analysis

Flow cytometric analysis of cell cycle distribution was completed as described previously (*Vanzyl et al., 2018*). Briefly, cells were treated with the indicated drugs for 24 h. For the final hour of drug treatment, the regular drug media was replaced with fresh media containing drug media supplemented with 30 μM 5′ Bromo- 2′ deoxyuridine (BrdU) (Sigma). After one hour, the media was removed, cells were washed with PBS and collected using trypsin and centrifugation. The pellet was washed with PBS and fixed in 70% ice cold ethanol for a minimum of 30 min at −20 °C. Following fixation, cells were pelleted and washed with PBS, then treated with 50 μg/ml RNAse A for 30 min at 37 °C. Samples were centrifuged, and cells were resuspended in 0.1M HCL 0.7% Triton X-100 and incubated on ice for 15 min. Cells were pelleted, resuspended in $H_2O$, and boiled for 15 min, then placed on ice for 15 min. 0.5% Tween 20 in PBS was added to the cells and then spun to pellet. The cell pellet was resuspended in Alexa Fluor 488-conjugated anti-BrdU antibody (BD Biosciences) in PBS, 5% FBS and 0.5% Tween 20 for 30 min in

the dark at room temperature. The antibody was removed, and cells were resuspended in 30 µM PI with 50 µM RNAse A in PBS. Samples were stained at 4 °C for a minimum of 30 min. BrdU incorporation and DNA content were measured using a BD Accuri C6 flow cytometer using the FL1 and FL2 channels, respectively, with BD Accuri C6 software.

### Sub G$_1$ assay

Flow cytometric analysis of sub G$_1$ DNA content was completed as described previously (*Vanzyl et al., 2018*; *Vanzyl et al., 2020*). Floating and adherent cells were collected and fixed in ice cold 70% ethanol for a minimum of 30 min at −20 °C. After fixation, cell pellets were washed twice with PBS and incubated in 30 µM PI with 50 µM RNAse A in PBS at 4 °C for 30 min. DNA content was measured using a BD Accuri C6 flow cytometer using the FL2 channel and analyzed using the BD Accuri C6 software.

## RESULTS

### The effect of ATF4 on Tg-induced changes in cell cycle distribution

ER stress and ATF4 overexpression led to G$_1$ arrest in a variety of cell types (*Frank et al., 2010*). However, the contribution of ATF4 to ER stress-induced alterations in cell cycle distribution had not been tested. To address this, we compared the effects of Tg on cell cycle distribution using HCT116 colon cancer cells and an ATF4-def subline that we generated recently (*van Zyl et al., 2021*) (Fig. 1A). We found that Tg exposure greatly increased ATF4 expression in parental HCT116 cells but that ATF4 was not detected before or after exposure to Tg in the isogenic ATF4-def subline (Fig. 1A). To assess the effects of Tg on cell cycle distribution, we used two-parameter flow cytometric analysis of BrdU incorporation *versus* DNA content. BrdU is an analog of thymidine, so it is incorporated into DNA during replication readily identifying cells in S phase during the BrdU labeling period. DNA content was estimated using PI. Together, this method is far more precise than estimates of cell cycle phase based on DNA content alone.

In the parental HCT116 cells, Tg led to the virtual depletion of S phase (Fig. 1B upper panels). The only BrdU positive cells remaining had close to 4 C DNA content and BrdU incorporation per cell was lower than the bulk of the S phase population in control samples, indicating that there was a small population of cells in late S phase near the S/G$_2$ boundary that incorporated some BrdU. Very few BrdU-positive cells were found in early S phase (Fig. 1B). There was also no significant accumulation of BrdU-negative cells with S phase DNA content (*i.e.,* no population between G$_1$ and G$_2$/M). Quantitative analysis of several independent experiments indicates that the proportion of parental cells in G$_1$ increased substantially with a corresponding decrease in S phase (Fig. 1C). There was a slight increase in the proportion of cells in the G$_2$ phase following Tg treatment. These alterations indicate that Tg induced a strong G$_1$ arrest and a less prominent G$_2$ arrest.

Cell cycle analysis in the ATF4 deficient subline was performed in parallel. There were no differences in cell cycle distribution prior to Tg treatment (Fig. 1B). Like controls, we detected a large decrease in S phase and a substantial increase in the G$_1$ phase of the cell cycle that were virtually identical to the controls (Fig. 1B). When quantified over multiple experiments, there were small but significant differences in G$_1$ and S phases between cell

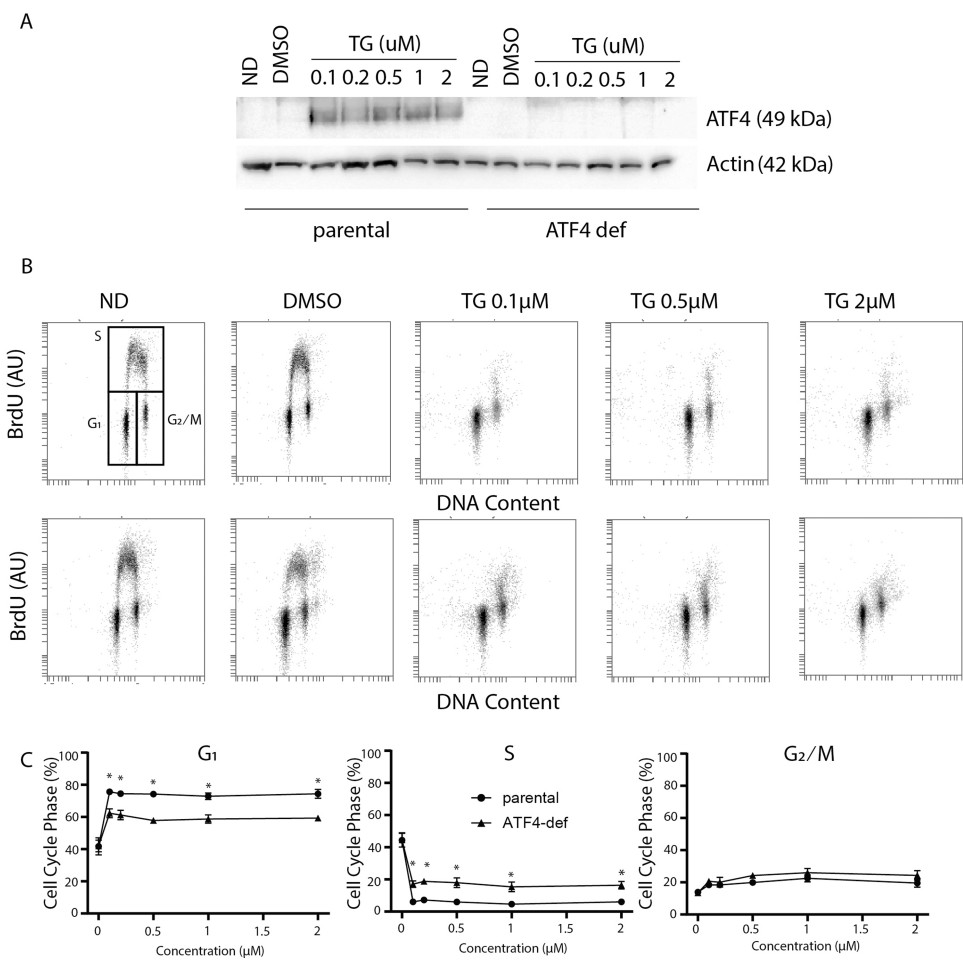

**Figure 1** **Disruption of ATF4 has relatively minor effects on thapsigargin-induced cell cycle alterations.** HCT116 parental and ATF4-def cells were treated with the indicated concentration of Tg for 24 h. In (A) and (B), ND and DMSO refer to no drug and vehicle controls, respectively. (A) Protein was extracted and immunoblot analysis of ATF4 and $\beta$-actin levels was performed. (B, C) One hour prior to collection, cells were labelled with 30 $\mu$M BrdU. Cells were fixed then stained with PI and anti-BrdU. Cells were read using the BD Accuri C6 flow cytometer using the FL-2 channel (PI) and FL-1 channel (anti-BrdU). (B) Representative dot plots of two parameter flow cytometric analysis of DNA replication (BrdU incorporation) *versus* DNA content (PI stain) are presented for parental HCT116 and ATF4-def cells (top and bottom, respectively). (C) Quantification of cell cycle data derived from dot plots like those in (B) were combined from three independent experiments. Each point represents the mean ($+/-$ SEM). Asterisks (* and **) indicate that the mean was significantly different from the corresponding value in the control cell line at $P < 0.05$ and $P < 0.01$, respectively, using two-way ANOVA followed by Tukey's multiple comparisons test.

lines following Tg treatment (Fig. 1C lower panels). Again, the few cells in S phase were in late S phase near S/$G_2$ boundary (Fig. 1C). The absence of BrdU positive cells with DNA content covering the spectrum from $G_1$ to $G_2$ indicates that Tg-induces $G_1$ arrest in the ATF4-deficient cells as well. The BrdU positive population of cells are likely those that entered S phase prior to the complete $G_1$ arrest and just reached the S/ $G_2$ boundary during the labeling period. There was a small but statistically significant increase in the proportion

of cells incorporating BrdU in the ATF4-def cell line. However, ATF4-def cells accumulated in $G_1$ phase indicating that ATF4 contributed little to the Tg-induced $G_1$ arrest.

## The effect of p53 on Tg-induced changes in cell cycle distribution

The p53 protein is a DNA damage-responsive transcription factor that positively regulates the cyclin-dependent kinase inhibitor p21$^{WAF1}$ leading to both $G_1$ and $G_2$ arrests (*Bunz et al., 1998*; *el Deiry et al., 1993*; *Xiong et al., 1993*; *Polyak et al., 1996*). In addition, exposure to Tg reportedly induces a PERK-dependent $G_2$ arrest through the upregulation of a specific isoform of p53 lacking the N-terminal transactivation domain (*Bourougaa et al., 2010*). Therefore, we sought to determine whether p53 contributed to Tg-induced changes in cell cycle distribution. Immunoblot analysis confirmed that ATF4 could be induced by Tg in the p53-null cells (Fig. 2A). We again used two-parameter flow cytometric analysis to determine the effects of Tg on cell cycle distribution in HCT116 and isogenic p53-null cells. There was no significant difference in cell cycle distribution before treatment (Fig. 2B). Again, Tg exposure led to an increase in $G_1$ and decrease in S phase in both cell lines comparable to the ATF4 parental HCT116 cells in Fig. 1 (Fig. 2B). The Tg treatment led to a large increase in $G_1$ along with a concomitant decrease in S phase across all concentrations (0.1–2 $\mu$M). Therefore, loss of p53 had no significant effect on Tg-induced ATF4 expression or Tg-induced cell cycle alterations.

## The effect of Tg on cell cycle distribution is largely p21$^{WAF1}$-dependent

The p21$^{WAF1}$ cyclin-dependent kinase inhibitor plays important roles in both $G_1$ and $G_2$ arrest induced by p53 (*Bunz et al., 1998*; *el Deiry et al., 1993*; *Xiong et al., 1993*; *Polyak et al., 1996*). In addition, it has been reported that ATF4 can regulate p21$^{WAF1}$ directly in response to ER stress (*Inoue et al., 2017*). Conversely, ER stress has also been reported to down regulate p21$^{WAF1}$ through the ATF4-regulated CHOP protein (*Mihailidou et al., 2010*). Therefore, we sought to determine if p21$^{WAF1}$ plays a role in Tg-induced alterations in cell cycle distribution. Immunoblot analysis HCT116 and an isogenic p21-null cell line indicated that p21$^{WAF1}$ was not detectable in p21-null cells even when p21$^{WAF1}$ was induced in the parental strain with the proteasome inhibitor MG132 (Fig. 3A). We went on to treat HCT116 and the p21$^{WAF1}$ null cells for 24 h with Tg. Again, the BrdU-positive S phase population of control cells was depleted in response to Tg (Fig. 3B). Remarkably, the Tg-treated p21$^{WAF1}$ null cells incorporated BrdU across S phase from the $G_1$/S to the S/$G_2$ boundary (Fig. 3B). Quantitative analysis of multiple experiments indicated that the Tg-induced $G_1$ arrest was abrogated in p21$^{WAF1}$ null cells and that the proportion of cells in S phase remained close to pre-treatment levels (Fig. 3C). The slight increase in $G_2$/M in the control cells was not detected in the p21$^{WAF1}$ null cells either. These results suggest that the $G_1$ and $G_2$ arrests resulting from Tg exposure were dependent on the p21$^{WAF1}$ protein.

As described above, ATF4 and p53 can both regulate p21$^{WAF1}$ expression, therefore it was important to assess the effect of Tg on CDKN1A mRNA and p21$^{WAF1}$ protein expression in these cell lines. We did detect a small but significant increase in CDKN1A mRNA at 24 h (Figs. 4A and 4B). However, immunoblot analysis indicates that p21$^{WAF1}$ levels were not significantly altered by Tg in control cells at 8 or 24 h (Figs. 4C–4F). The

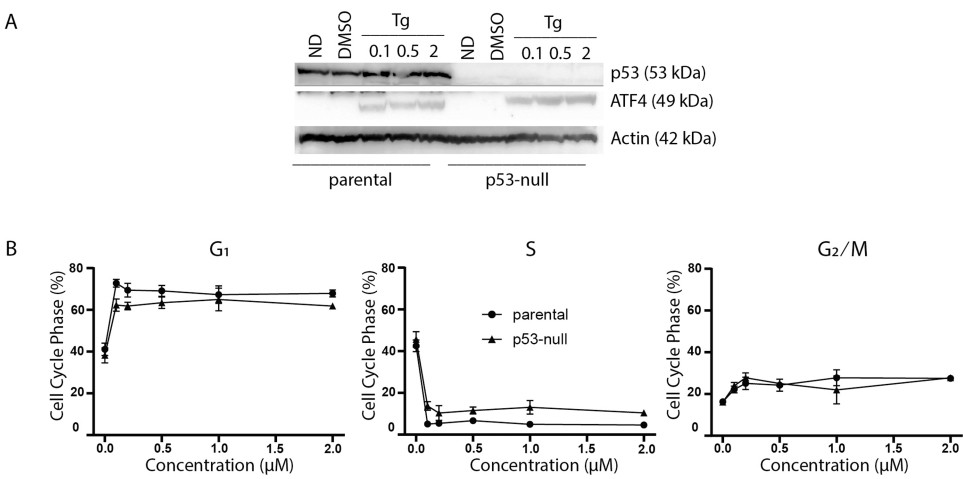

**Figure 2  P53 does not contribute to thapsigargin-induced cell cycle alterations.** HCT116 parental and the p53-null subline were treated with between 0 and 2 µM Tg 24 h. (A) Protein was extracted and immunoblot analysis of p53 and ATF4 levels was completed. ND and DMSO refer to no drug and vehicle controls, respectively. (B) One hour prior to collection cells were labelled with 30 µM BrdU for one hour. Cells were fixed then stained with PI and anti-BrdU. Cells were read using the BD Accuri C6 flow cytometer using the FL-2 channel (PI) and FL-1 channel (anti-BrdU). Quantification of cell cycle data derived from dot plots like those in (Fig. 1B) were combined from three independent experiments. Each point represents the mean (+/− SEM). An asterisk (*) indicates that the mean was significantly different from the corresponding value in the control cell line at $P < 0.05$, using two-way ANOVA followed by Tukey's multiple comparisons test.

basal expression of p21[WAF1] was lower in the p53-null cells, as expected (*Kaeser & Iggo, 2002*; *Tang et al., 1998*), and wasn't increased by Tg (Figs. 4C and 4E). The constitutive expression of p21[WAF1] in the ATF4-def line was a little higher than isogenic controls and again Tg had no additional effect on p21[WAF1] levels in the ATF4-def cells (Fig. 4D and 4F). So, there was discordance between changes in CDKN1A mRNA and p21[WAF1] protein levels. This is likely explained by the fact that Tg inhibits protein synthesis (*van Zyl et al., 2021*; *Wong et al., 1993*) through the PERK-dependent branch of the UPR (*Hetz, 2012*). The small increase in mRNA was unlikely to lead to a comparable increase in the encoded protein. Taken together, p21[WAF1] plays a key role in determining the effect of Tg on cell cycle progression but p21[WAF1] levels themselves were unaltered by Tg exposure.

## p21[WAF1], *p53* and *ATF4*-deficient cells are hypersensitive to Tg-induced apoptosis

Tg is a well-known inducer of apoptosis (*Tsukamoto, Kaneko & Kurokawa, 1993*; *Jiang et al., 1994*). This can occur through an ATF4-dependent manner through UPR activation and the upregulation of pro-apoptotic proteins including CHOP (*Sehgal et al., 2017*; *Lindner et al., 2020*; *Teske et al., 2013*; *Matsumoto et al., 2013*). Therefore, we sought to determine how ATF4, p53 and p21[WAF1] disruption affected sensitivity of HCT116 cells to Tg. We used one parameter flow cytometric analysis of PI-stained cells to estimate sub $G_1$ DNA content. Unexpectedly we found that the ATF4-def cells were more sensitive to Tg treatment than their parental controls across all concentrations of Tg (Fig. 5A). Similarly, we found that

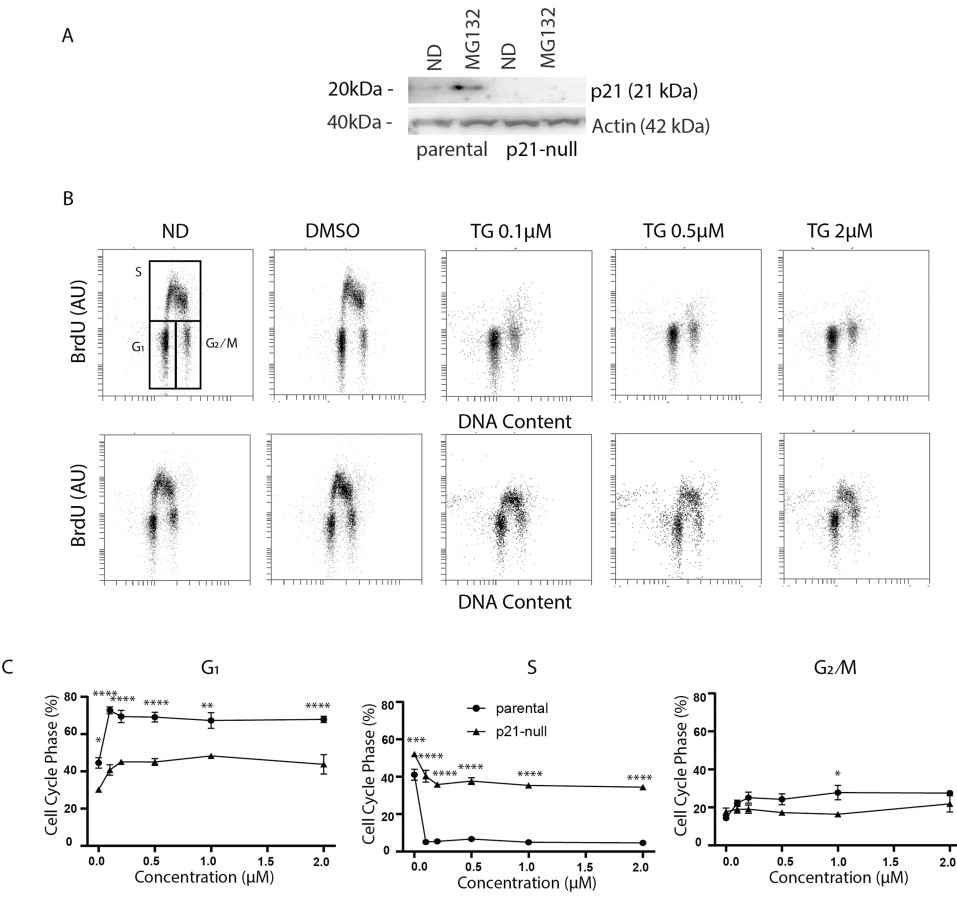

**Figure 3** **Tg induces a $G_1$ arrest dependent on p21$^{WAF1}$.** HCT116 parental and the p21-null subline were treated with between 0 and 2 μM Tg 24 h. In (A) and (B), ND and DMSO refer to no drug and vehicle controls, respectively. (A) Where indicated, cells were treated with 10 μM MG132 for 8 h and immunoblot analysis was performed to confirm that p21-null cells didn't express p21$^{WAF1}$. (B and C) Cells were treated with up to 2 μM Tg for 24 h. One hour prior to collection, 30 μM BrdU was added to cells. Cells were collected, fixed then stained with PI and anti-BrdU (BD). Cells were read using the BD Accuri C6 flow cytometer using the FL-2 channel (PI) and FL-1 channel (anti-BrdU). (B) Representative dot plots are shown for HCT parental and the p21-null subline (top and bottom, respectively). (C) Represents the quantification of the proportion of cells in each phase of the cell cycle from 3 experiments like those presented in (B). Each point represents the mean ($+/-$ SEM). Asterisks (*, **, *** and ****) indicate that the mean was significantly different from the corresponding value in the control cell line at $P < 0.05$, 0.01, 0.005 and 0.0001, respectively, using two-way ANOVA followed by Tukey's multiple comparisons test.

both the p21-null and p53-null cells were more sensitive to Tg treatment than their parental cells (Figs. 5B and 5C). The sub-$G_1$ apoptosis assay assesses the proportion of cells with fragmented DNA (*Nicoletti et al., 1991*). To ensure that the large increase in the proportion of cells with sub-G1 DNA was associated with apoptosis, we treated HCT116 cells and the p21-null cell line with Tg in the presence of the broad range caspase inhibitor (zVAD-fmk). This caspase inhibitor effectively blocked the Tg-induced increase in the sub-$G_1$ population of cells (Fig. 6). Therefore, the generation of the sub-G1 population was caspase dependent. These results indicate that ATF4, p21$^{WAF1}$ and p53 are acting in a protective manner to

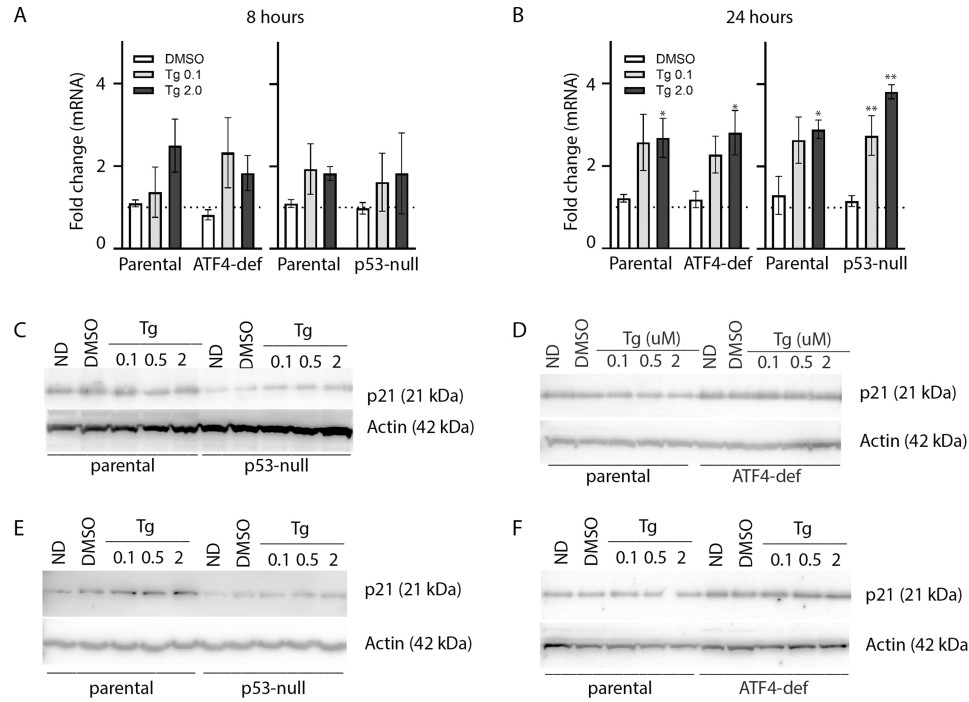

**Figure 4** **Tg induces CDKN1A mRNA in an ATF4- and p53-independent manner without an increase in p21^WAF1 protein.** HCT116 parental, p53-null and ATF4-def sublines were treated with up to 2 μM Tg for either 8 h (A, C and D) or 24 h (B, E and F). (A and B) RNA was extracted, reverse transcribed and used to assess CDKN1A mRNA levels using qRT-PCR. Data was normalized to GAPDH and then no drug control samples and expressed as the mean fold change determined from between 3 and 6 independent experiments. Asterisks (* and **) indicate that the value were significantly different from DMSO controls at $P < 0.05$ and $P < 0.001$, respectively, by one-way ANOVA followed by Dunnett's multiple comparisons test. (C–F) Total protein was isolated for immunoblot analysis of p21^WAF1 protein levels. ND and DMSO were no drug and vehicle controls, respectively. Immunoblots presented are representative of three independent biological experiments.

prevent Tg-induced apoptosis in HCT116 cells. This protective effect does not appear to be related to the differences in cell cycle regulation detected in this isogenic series of cell lines.

## The effect of Tg on cell cycle distribution and apoptosis in human fibroblasts

The results above were all generated in HCT116 cells and genetically modified sublines. It was important to determine if the response of Tg-treated HCT116 cells were unique or common to other cell types. Exposure to Tg similarly caused non-transformed human fibroblasts (NFhTert cells) to arrest predominantly in the $G_1$ phase of the cell cycle with a concomitant decrease in S phase across all drug concentrations (Fig. 7A). Furthermore, the proportion of NFhTert cells undergoing Tg-induced apoptosis was similar to HCT116 cells (Fig. 7B) and could be blocked with zVAD-fmk (Fig. 7C). Therefore, the responses of HCT116 and NFhTert cells to Tg across all drug concentrations were similar.

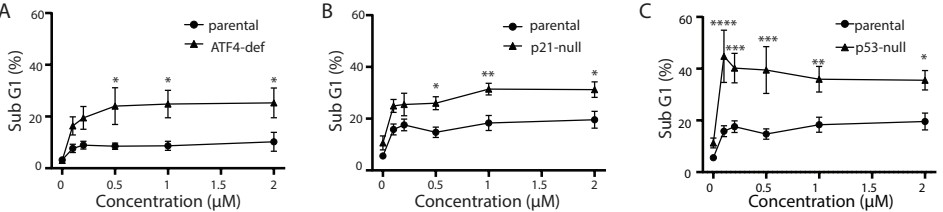

**Figure 5** **ATF4-, p53- and p21[WAF1]-deficient cell lines are more sensitive to Tg.** HCT116 parental, ATF4-def (A), p21-null (B) and p53-null (C) sublines were treated with up to 2 μM Tg, or an equivalent volume of DMSO, for 48 h. Both adherent and floating cells were collected, fixed and stained with PI to determine the proportion of cells with sub G1 DNA content. Each point represents the mean of between three and six independent experiments. Error bars represent the SEM, and a two-way ANOVA was used to determine significant differences between the cell lines where *, **, *** and **** indicate that $P < 0.05$, 0.01, 0.005 and 0.0001, respectively.

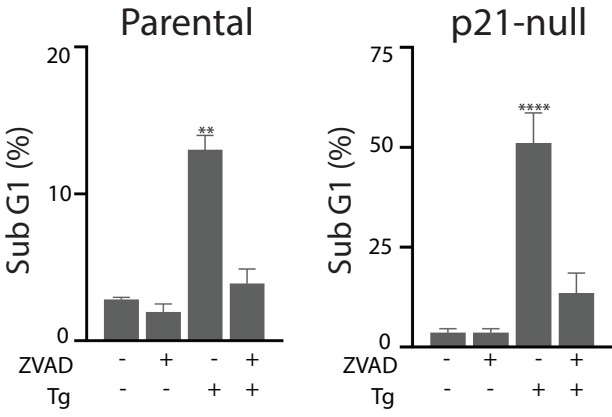

**Figure 6** **Tg-induced cell death is inhibited by zVAD-fmk.** HCT116 cells and the p21-null subline were treated with Tg, zVAD-fmk or both compounds for 48 h and the proportion of cells with subG1 DNA content was determined by flow cytometry. Each value represents the mean +/− SEM determined from three independent experiments. Asterisks (** and ****) indicate that the value is significantly different from the DMSO control ($P < 0.01$ and $P < 0.0001$) by one-way ANOVA followed by a Dunnett multiple comparisons test.

## DISCUSSION

Tg is a competitive inhibitor of SERCA, so Tg interferes with Ca2+ uptake to the lumen of the ER (*Sabala et al., 1993*; *Inesi & Sagara, 1992*). The decrease in luminal Ca2+ reduces chaperone activity leading to the accumulation of misfolded proteins (*Coe & Michalak, 2009*). This ER stress leads to the activation of the UPR, a series of signaling cascades initiated in the ER that ultimately reduce protein load in the ER to restore homeostasis (*Hetz, 2012*). The three branches of the UPR are initiated through three separate ER-resident transmembrane proteins (Ire1, PERK and ATF6) that through distinct mechanisms lead to the activation of specific transcription factors (XBP1s, ATF4 and ATF6) (reviewed in *Hetz*

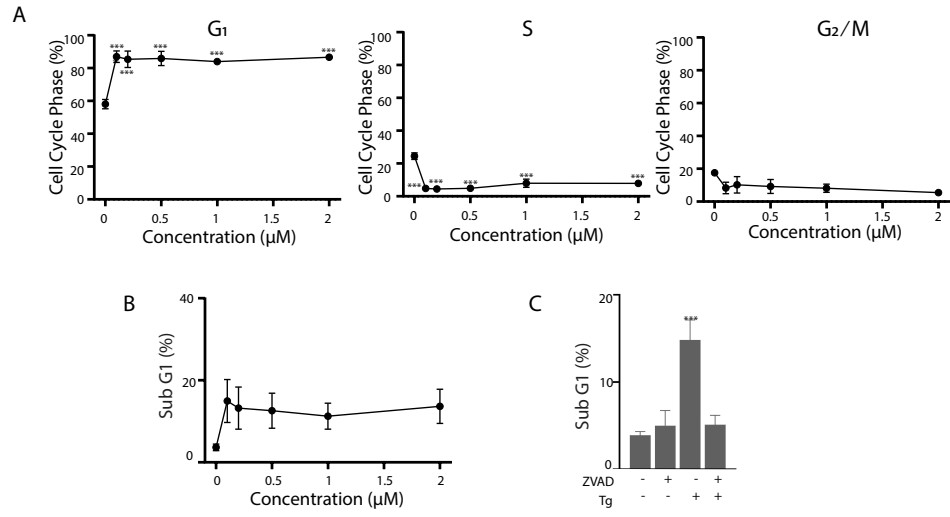

**Figure 7** **Tg induces G₁ cell cycle arrest and apoptosis in human fibroblasts.** (A and B) NFhTERT cells were treated with up to 2 μM Tg and cells were collected at 24 h for cell cycle analysis (A) and 48 h for apoptosis (B). (C) NFhTert cells were treated with Tg, zVAD-fmk or both compounds for 48 h and the proportion of cells undergoing apoptosis was determined by flow cytometry. Each value in (A–C) represents the mean +/− SEM determined from 3–6 independent experiments. Asterisks (***) indicate that the value is significantly different from the DMSO control ($P < 0.001$) by one-way ANOVA followed by a Dunnett comparisons test.

*& Papa, 2018*). XBP1s and ATF6 increase the expression of ER chaperones to facilitate protein folding in the ER while XBP1s also promotes the removal and destruction of misfolded proteins through the upregulation of proteins involved in the endoplasmic-reticulum-associated protein degradation pathway (ERAD) (reviewed in *Hetz & Papa, 2018*). The PERK-dependent branch of the UPR is less specific to ER stress because it is part of the ISR that also responds to dsRNA, heme deficiency and amino acid deprivation through the PKR, HRI and GCN1 kinases (*Pakos-Zebrucka et al., 2016*). All 4 active kinases phosphorylate eIF2α leading to decreased protein synthesis while permitting the selective translation of the ATF4 transcription factor with consequent increases in ATF4-responsive gene expression (*Pakos-Zebrucka et al., 2016*). Therefore, Tg increases the expression of ATF4-regulated proteins common to other ISR pathways so this part of the UPR is a more general stress response.

The 78 kDa glucose regulated protein (GRP78), also known as BiP, is an ER chaperone that controls the UPR (reviewed in *Lee, 2005*). In unstressed conditions, GRP78 binds to the luminal domains of the transmembrane sensors (PERK, Ire1 and ATF6) blocking their roles in ER signaling (*Bertolotti et al., 2000*). Focusing on the PERK pathway specifically, the protein homodimerizes and its cytoplasmic domain is autophorylated (*Bertolotti et al., 2000*). Activated PERK can phosphorylate eIF2α to inhibit translation. Under these conditions, ATF4 is selectively translated through a mechanism involving upstream open reading frames in its 5′ UTR (*Vattem & Wek, 2004*). The ATF4 protein is translocated to the nucleus where it regulates the expression of many genes (manuscript in progress).

These include DDIT3 and HSP5A (encoding CHOP and GRP78) that play pro- and anti-apoptotic roles in response to ER stress, respectively.

As outlined earlier, Tg induces PERK-dependent alterations in the cell cycle (*Han et al., 2013*; *Hamanaka et al., 2005*; *Cabrera et al., 2017*; *Popat, Patel & Warnes, 2019*). It has been reported that DNA synthesis is inhibited within 1 h in Tg leading to decreased incorporation of BrdU, consistent with a cell cycle arrest in S phase (*Cabrera et al., 2017*). Our data on a longer time scale suggests that this is not sustained because we only detected a relatively small decrease in BrdU incorporation per cell in the p21-null cells. Instead, we detected a decrease in the proportion of cells incorporating BrdU in all other cell lines. This is consistent with $G_1$ not S phase arrest. Other investigators have reported that Tg-induced $G_1$ arrest is PERK- and GCN2-dependent following similar long exposure to ER stressors (*Hamanaka et al., 2005*). However, previous work didn't assess the role of downstream targets in the ISR. These ISR pathways converge on ATF4 because ATF4 is downstream of both PERK and GCN2 (*Pakos-Zebrucka et al., 2016*). It is also noteworthy that forced expression of ATF4 can also lead to $G_1$ arrest (*Frank et al., 2010*). However, the present work suggests that ATF4 is not essential for Tg-induced $G_1$ arrest because a sizable $G_1$ arrest was detected in ATF4-deficient cells. Nonetheless, there was a small but significant decrease in the proportion of ATF4-deficient cells in the $G_1$ phase of cell cycle following Tg treatment.

$G_2$ arrest is reported to be induced by Tg in a PERK-dependent manner but it appears to also require an isoform of the p53 tumour suppressor that lacks its N-terminal transactivation domain (*Bourougaa et al., 2010*). In our hands, Tg led to a small increase in the proportion of cells in $G_2$/M but deletion of p53 in HCT116 cells (*Bunz et al., 1998*) didn't affect the accumulation of cells with 4C DNA content suggesting that this reported mechanism was not contributing to arrest. Similarly, ATF4-deficient cells did not have a grossly altered $G_2$ arrest. The most striking cell cycle alteration in control, p53-null and ATF4-deficient cells was the complete absence of a replicating population of cells in early S phase, indicative of a complete block to S phase entry. Therefore, there is not a requirement for p53, N-terminal deleted variants of p53 or ATF4 in the $G_1$ arrest detected here.

Both ATF4 and p53 can regulate expression of the cyclin-dependent kinase inhibitor p21$^{WAF1}$ an important $G_1$ and $G_2$ checkpoint regulator (*Inoue et al., 2017*; *Ebert et al., 2020*; *Lehman et al., 2015*). The p21$^{WAF1}$ protein is a negative regulator of cell cycle progression because it inhibits cyclin D/cdk4, cyclin E/cdk2, cyclin A/cdk2, cyclin A/cdk1 and cyclin B/cdk1 activity important for progression through $G_1$, S, $G_2$ and M phases of the cell cycle (*Harper et al., 1993*; *Abbas & Dutta, 2009*). In our experiments, the p21-null cells continued to enter S phase and synthesize DNA while the parental controls arrested in $G_1$ and to a lesser extent $G_2$. Our results indicate that the $G_1$ arrest induced by Tg is dependent on p21$^{WAF1}$ and that p21$^{WAF1}$ also contributes to Tg-induced $G_2$ arrest. However, this appears to be independent of p53 and ATF4 and does not require an increase in p21$^{WAF1}$ levels.

The fact that p21$^{WAF1}$ levels did not increase under conditions that induced $G_1$ arrest is somewhat surprising. ATF4 was upregulated in our experiments and ATF4 can directly regulate the CDKN1A gene encoding p21$^{WAF1}$ (*Inoue et al., 2017*). However, ER stress

has been reported to decrease p21$^{WAF1}$ levels through a CHOP (DDIT3) dependent mechanism (*Mihailidou et al., 2010*). CHOP is downstream of ATF4 in the ISR and PERK-dependent branch of the UPR (*Hetz & Papa, 2018*) so activation of these pathways can affect p21$^{WAF1}$ expression in different ways. The data presented here indicates that CDKN1A mRNA levels increased significantly but a change in p21$^{WAF1}$ protein level was not detected. This discordance between mRNA and protein is not entirely surprising because one of the adaptive mechanisms that protects cells from ER stress is through the inhibition of protein synthesis to reduce the burden of misfolded proteins in the ER (*Hetz & Papa, 2018*). This occurs through the phosphorylation of the translation initiation factor eIF2 α by the four eIF2 α kinases of the ISR and UPR (*Taniuchi et al., 2016*; *Donnelly et al., 2013*). In our hands, Tg can reduce protein synthesis by 90% in these cell lines (*van Zyl et al., 2021*). The efficient synthesis of any protein under these conditions would require a mechanism to circumvent the decrease in the rate of translation. Although the mechanism linking p21$^{WAF1}$ to the Tg-induced G$_1$ arrest has not been fully elucidated, it is clear that the basal levels of p21$^{WAF1}$ were sufficient for this arrest.

Apoptosis can be readily induced by Tg in a variety of cell types (*Tsukamoto, Kaneko & Kurokawa, 1993*; *Jiang et al., 1994*). This can occur through the PERK-dependent branch of the UPR, involving ATF4 and ATF4-regulated pro-apoptotic proteins (*Sehgal et al., 2017*; *Lindner et al., 2020*; *Teske et al., 2013*; *Matsumoto et al., 2013*). In the present work, we found that loss of ATF4 increased sensitivity to apoptosis suggesting that the PERK-eIF2 α-ATF4 pathway of the UPR is functioning in an adaptive manner in HCT116 cells across the broad range of Tg concentrations tested. Deletion of TP53 and CDKN1A was also protective against apoptosis. This protection does not appear to be related to the G$_1$ arrest because only p21$^{WAF1}$ contributed convincingly to G$_1$ arrest, but all three proteins protected these cells from ER stress.

## CONCLUSIONS

ER stress can lead to induction of ATF4 through the PERK-dependent branch of the UPR that converges on a variety of stress responses collectively known as the integrated stress response. It has been previously shown that ER stress causes alterations in cell cycle distribution, however the role of ATF4 in these alterations had not been examined genetically. Here, we showed that Tg contributes to a pronounced G$_1$ arrest but that this is not dependent on ATF4 or p53 but appears to require the p21$^{WAF1}$ cyclin-dependent kinase inhibitor. Although ATF4 can reportedly upregulate p21$^{WAF1}$, the evidence here suggests that p21$^{WAF1}$ levels do not increase in response to Tg. The important role uncovered here for p21$^{WAF1}$ suggests that inhibition of G$_1$ cyclin-dependent kinase activity is important in Tg-induced cell cycle arrest.

## ACKNOWLEDGEMENTS

We would like to thank Bert Vogelstein for providing p53- and p21-null cells, along with their control HCT116 cells. We would also like to thank Mats Ljungman for providing the NFhTert cells.

### Funding

This work was supported by the Natural Sciences and Engineering Research Council of Canada (RGPIN-2019-06146). Erin van Zyl was supported with a Vanier Canada Graduate Scholarship through the Natural Sciences and Engineering Research Council of Canada. Erin van Zyl and Claire Peneycad received studentship support through the Ontario Graduate Scholarship program from the Ontario Ministry of Colleges and Universities. The funders had no role in study design, data collection and analysis, decision to publish, or preparation of the manuscript.

### Grant Disclosures

The following grant information was disclosed by the authors:
Natural Sciences and Engineering Research Council of Canada: RGPIN-2019-06146.
Vanier Canada Graduate Scholarship through the Natural Sciences and Engineering Research Council of Canada.
Ontario Graduate Scholarship program from the Ontario Ministry of Colleges and Universities.

### Competing Interests

The authors declare there are no competing interests.

### Author Contributions

- Erin van Zyl conceived and designed the experiments, performed the experiments, analyzed the data, prepared figures and/or tables, authored or reviewed drafts of the article, and approved the final draft.
- Claire Peneycad conceived and designed the experiments, performed the experiments, analyzed the data, prepared figures and/or tables, and approved the final draft.
- Evan Perehiniak conceived and designed the experiments, performed the experiments, analyzed the data, prepared figures and/or tables, and approved the final draft.
- Bruce C. McKay conceived and designed the experiments, analyzed the data, prepared figures and/or tables, authored or reviewed drafts of the article, and approved the final draft.

### Data Availability

The uncropped immunoblots are available in the Supplementary Files.

### Supplemental Information

Supplemental information for this article can be found online at http://dx.doi.org/10.7717/peerj.16683#supplemental-information.

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
