# Peer review of "Cyclin-dependent kinase inhibitor 1 plays a more prominent role than activating transcription factor 4 or the p53 tumour suppressor in thapsigargin-induced G1 arrest"

_PeerJ, doi:10.7717/peerj.16683_

## Round 0.1 · original submission · Major Revisions

Please respond to the reviewers point by point.

·

Basic reporting

The article was professionally written, ensuring ease of comprehension and a clear, logical flow that made it easy to follow.

Experimental design

This study is well-designed.

Validity of the findings

The results presented in this study successfully address the core research question.

Additional comments

Minor concerns:
1. The title of the article is overly definitive and not appropriate. It should be revised to reflect that p21 is a key gene influencing G1 arrest induced by Tg. Additionally, the statistical data in Figures 1 and 2 suggest that ATF4 and P53 also have effects on G1 arrest.
2. For all Western blots, it is recommended to use an inverted display.
3. For all the FACS data, it is recommended to gate different cell cycle phases and show the percentage of cells in each phase.
4. In Figure 1, HCT116 should be labeled as HCT116-WT or HCT116-Control.
5. In Figure 4A, please remove the line around Actin, and in Figure 4B, remove the letter under "ND".
6. Line 252 should be modified to include "(Figure 5)" at the end of the sentence: "Tg treatment showed higher expression levels compared to their parental cells (Figure 5)."
7. In Figure 5, it would be better to include FACS data and consider including apoptosis marker staining in cells or performing a Western blot to check for apoptosis markers like cleaved caspase-3.

Reviewer 2 ·

Basic reporting

The manuscript is well written and clear to read.
Literature references and background is sound.
The figures can be improved to better professional standards.
1. For improved data presentation, the authors are suggested to invert the western blot images i.e. show black bands on a white background.
2. The authors are suggested to add molecular weights to the protein ladder for western blots.
3. Additionally, indicate ATF4 with an arrow.
4. Designate the usage of * in figure legend.
5. Expand ND in figures in the legend.

Experimental design

While the authors use HCT116 for the availability of isogenic cell lines, the Thapsigargin response as indicated by cell death data in the subG1 experiment in Figure 5 suggests a saturating dose all the way from 0.25 - 2uM concentrations. These cells only exhibit 20% cell death in the control cells at doses used in the manuscript. This suggests HCT116 is resistant to Tg induced cell death, at least at the 48 hour timepoint displayed in the figures. This is not a typical response of most cell lines and hence it will not be a representative cell line to comment of p21 and its effect of UPR more broadly.
Kindly address these comments.

Validity of the findings

With Reference to Figure 4, there is contradictory results in Inoue et al publication (also cited by authors) wherein they report p21 induction in HCT116 in the 8 hour time line.
It is recommended to do a detailed Thapsigargin time course with several time points between 2- 24 hours for a rigorous experiment.
Maybe RNA level of p21 may be a more sensitive measure of detection in a time course experiment.

Reviewer 3 ·

Basic reporting

Adaptive response to Tg treatment, or ER stress in general, included transient growth arrest which may be regulated at multiple level including down regulation of protein synthesis. Multifaced signaling cascade are induced during ER stress. Here Zyl et.al. investigated the role of ATF4, p53 and p21 in Tg-induced cell cycle arrest and cell death. ATF4, p53 and p21 knocked out isogenic HCT116 cell lines were used to test the role of these three genes on Tg induced cell cycle arrest and cell death. Cell cycle analysis showed modest effect of ATF4 deletion on Tg induced G1 arrest and concomitant increase in S while p53 deletion did not change the Tg effect on cell cycle. P21 knockout cell line showed decrease in both basal levels well as Tg-induced in G1 population. Concomitantly p21 knockout cell maintains cycling cell population after 24h Tg treatment. Despite differential effect on cell cycle, all the three genes are required for cell survival under Tg treatment as indicated by increased cell death in KO lines. Overall results are interesting and relevant for cellular adaptation to ER stress. However, all the experiments were performed in only one cell line which limits the broader scope of the manuscript. At least some of the key observation should be tested on at least another cell line.
Specific comments:
1. At least some of the key observation should be tested on at least another cell line.
2. Figure 1c: There are significant difference in G1 and S phases between Tg-treated parental and ATF4-deficienct HCT116 cells. Therefore, the conclusion (line 199) should be revised to reflect the modest but significant changes in the G1 and S phases. Same comment for Figure 2b and line 214.
3. Figure 2a: Tg was shown to induced p53/47 alternative isoforms and G2 arrest was proposed to be dependent on the p53/47 (Ref#10). Do authors see such smaller size band in the full blot after Tg treatment? If yes, authors should check whether p53/47 induction depends on ATF4 (figure 1a blot). If not, the alternative translation initiation of p53 mRNA could be cell type dependent which should be discussed.
4. Figure 4a-d, line 240: If p21 expression is unaltered by Tg treatment, how does p21 function is altered to increased cell cycle arrest in parental cells?
5. Figure 5: sub G1 DNA content is a good approximation for cell death, however, an direct assay for cell death will strength the conclusion.
General formatting:
6. Figure 1a and other immunoblots: Immunoblots are conventionally represented as inverted images, for example, dark bands and light background.
7. Figure1b: Indicate representative gatings for G1, S and G on the scatter plots. In this analysis, G2 can’t be distinguished from M. Therefore, it should be labelled as G2/M.

Experimental design

Detailed in Basic reporting.

Validity of the findings

Detailed in Basic reporting.

Additional comments

Detailed in Basic reporting.

---

## Round 0.2 · Major Revisions

Please respond to the reviewers and address the comments point by point.

·

Basic reporting

no comment

Experimental design

no comment

Validity of the findings

no comment

Additional comments

The authors have adequately addressed my concerns. I still think that using an inverted display, as is common in most papers, would be better for Western blots.

Reviewer 2 ·

Basic reporting

See additional comments

Experimental design

See additional comments

Validity of the findings

See additional comments

Additional comments

The authors have addressed some of my concerns but not all . Below is the rebuttal along with my responses as Reviewer 2.
Reviewer 2
1. For improved data presentation, the authors are suggested to invert the western blot images i.e. show black bands on a white background.
See comment 2 to reviewer 1.

Reviewer 2 : It is a publication standard to display inverted images of western blots.

2. The authors are suggested to add molecular weights to the protein ladder for western blots.
We had indicated the molecular weight of each protein in western blots but we have now added the molecular weight of all bands on the full western blot images provided as supplementary data.

Reviewer 2: This change has been made to supplementary images of western blots.

3. Additionally, indicate ATF4 with an arrow.
I assume the reviewer is referring to the full images provided as supplementary data, not the cropped images in the body of the text. We have added arrows to the supplementary figures.
Reviewer 2: This had been addressed.
4. Designate the usage of * in figure legend.
This has been done.

Reviewer 2: Addressed

5. Expand ND in figures in the legend.
This has been done by including the following statement in relevant figure legends: ND and DMSO refer to no drug and vehicle controls, respectively.

Reviewer 2: Addressed satisfactorily.

6. While the authors use HCT116 for the availability of isogenic cell lines, the Thapsigargin response as indicated by cell death data in the subG1 experiment in Figure 5 suggests a saturating dose all the way from 0.25 - 2uM concentrations. These cells only exhibit 20% cell death in the control cells at doses used in the manuscript. This suggests HCT116 is resistant to Tg induced cell death, at least at the 48 hour timepoint displayed in the figures. This is not a typical response of most cell lines and hence it will not be a representative cell line to comment of p21 and its effect of UPR more broadly.
Kindly address these comments.
We appreciate these comments. It is possible that the percentage of cells undergoing apoptosis detected in this way and under these conditions is lower than other cell lines. In the revised manuscript, we chose to examine the Tg response in non-transformed neonatal fibroblasts, assuming that non-cancer cell lines would exhibit a “typical” response. The Tg-treated fibroblasts exhibited a similar saturating apoptosis and G1 arrest response to HCT116 cells. This data is now included as Figure 7. Hopefully this allays the concern of the reviewer.

Reviewer 2: The authors display data percentage for cells in sub G1 phase. If they instead display actual cell death counts data, that would address this point accurately. I see that Reviewer 1 has also suggested cell death analysis by FACS and/or western blotting in point#7.
In the revised manuscript, the authors have conducted experiments with the caspase inhibitor zVAD and again plotted percentage of cells in subG1 phases. If they instead show us numbers for the cell death assay, it would allay the concerns of Reviewer 1 and me.

7. With Reference to Figure 4, there is contradictory results in Inoue et al publication (also cited by authors) wherein they report p21 induction in HCT116 in the 8 hour time line.
It is recommended to do a detailed Thapsigargin time course with several time points between 2- 24 hours for a rigorous experiment.
Maybe RNA level of p21 may be a more sensitive measure of detection in a time course experiment.
Thank you for the comment. We had already measured CDKN1A mRNA in HCT116 cells following exposure to 2uM Tg at 2, 8 and 24 hours. We detected a small increase in CDKN1A RNA at 8 and 24 hours (2 to 2.5 fold) but there was no change at the protein level in response to TG. The small change in RNA level was not associated with increased p21 protein at any time. The difference in CDKN1A mRNA and p21 protein expression is interesting but we don’t know if the induced transcript is translated. It is ultimately the protein that is the functional gene product and p21 levels do not change under conditions in which the cells arrest in G1.

Reviewer 2 It will be useful to display this mRNA data and cite the paper above mentioned paper for accurate representation. Considering that this experiment in the same cell line and conditions has been previously published it is all the more important to address the point thoroughly with potential explanation in the current manuscript.

---

## Round 0.3 · accepted · Accept

All the comments were addressed by the authors.

·

Basic reporting

no comment

Experimental design

no comment

Validity of the findings

no comment

Additional comments

The substantially revised manuscript has adequately addressed all reviewer comments.